# Trends in gender and socioeconomic inequalities in mental health following the Great Recession and subsequent austerity policies: a repeat cross-sectional analysis of the Health Surveys for England

Rachel M Thomson,[1,2] Claire L Niedzwiedz,[3] Srinivasa Vittal Katikireddi[1]

[1]MRC/CSO Social & Public Health Sciences Unit, University of Glasgow, Glasgow, UK
[2]Public Health Department, NHS Ayrshire & Arran, Ayr, UK
[3]Institute of Health & Wellbeing, University of Glasgow, Glasgow, UK

**Correspondence to**
Dr Rachel M Thomson;
rachel.thomson@glasgow.ac.uk

## ABSTRACT

**Objective** It is known that mental health deteriorated following the 2008 global financial crisis, and that subsequent UK austerity policies post-2010 disproportionately impacted women and those in deprived areas. We aimed to assess whether gender and socioeconomic inequalities in poor mental health have changed since the onset of austerity policies.

**Design** Repeat cross-sectional analysis of survey data.

**Setting** England.

**Participants** Nationally and regionally representative samples of the working-age population (25–64 years) from the Health Survey for England (1991–2014).

**Outcome measures** Population-level poor mental health was measured by General Health Questionnaire-12 (GHQ) caseness, stratified by gender and socioeconomic position (area-level deprivation and highest educational attainment).

**Results** The prevalence of age-adjusted male GHQ caseness increased by 5.9% (95% CI 3.2% to 8.5%, p<0.001) from 2008 to 2009 in the immediate postrecession period, but recovered to prerecession levels after 2010. In women, there was little change in 2009 or 2010, but an increase of 3.0% (95% CI 1.0% to 5.1%, p=0.004) in 2012 compared with 2008 following the onset of austerity. Estimates were largely unchanged after further adjustment for socioeconomic position, employment status and household income as potential mediators. Relative socioeconomic inequalities in GHQ caseness narrowed from 2008 to 2010 immediately following the recession, with Relative Index of Inequality falling from 2.28 (95% CI 1.89 to 2.76, p<0.001) to 1.85 (95% CI 1.43 to 2.38, p<0.001), but returned to prerecession levels during austerity.

**Conclusions** Gender inequalities in poor mental health narrowed following the Great Recession but widened during austerity, creating the widest gender gap since 1994. Socioeconomic inequalities in poor mental health narrowed immediately postrecession, but this trend may now be reversing. Austerity policies could contribute to widening mental health inequalities.

## Strengths and limitations of this study

► Data are from a large nationally and regionally representative survey, and our study considered trends over a long period of time using a validated measure of poor mental health.
► Inequalities in poor mental health were explored by both socioeconomic position (using two measures to demonstrate consistency of trends) and gender, rarely explicitly done in current literature.
► Lack of available data meant it was not possible to categorise individuals according to whether they were subject to specific austerity measures; further research with such data and a clear control group would strengthen arguments for causality.
► The use of cross-sectional rather than longitudinal data mean the ability to derive causal inferences is limited and further longitudinal work is required.

## INTRODUCTION

The health and social repercussions of the 2008 Great Recession are still being felt today.[1 2] Much existing research has focused on the relationship between the economic downturn, rises in unemployment and worsening mental health outcomes.[3 4] Mirroring historical trends, in the aftermath of the recession there was an improvement in all-cause mortality across Europe,[5] paradoxically accompanied by a sharp rise in suicide rates which disproportionately impacted men.[6]

There has been a growing call to interpret trends in mental health outcomes in the context of the political decisions that followed,[7 8] particularly given that there was marked cross-national variation in these outcomes.[9] It has been argued that the pursuit of austerity policies in response to

the recession, usually involving large-scale public sector reforms, may actually have worsened health outcomes and delayed economic recovery.[10–12] It has also been postulated that austerity policies may worsen inequalities in health outcomes, as they frequently result in cutbacks to programmes aiming to address inequitable distribution of the social determinants of health such as housing and education.[13]

The package of austerity measures implemented by the UK government in 2010 was the third largest in Europe, with substantial cuts especially to welfare, health and social care.[14] Between 2010 and 2015, £26 billion worth of cuts were made to benefits, tax credits, pay and pensions in the UK,[15] with local authorities serving more deprived communities seeing greater financial losses.[16] Eighty-five per cent of financial savings from welfare reforms have been taken from the incomes of women, largely due to the fact that they make up the majority of lone parents and unpaid carers.[17] Women also form a large proportion of the public sector workforce, two-thirds in 2012–2013,[18] so are more likely to have been impacted by the 2-year public sector pay freeze in 2010 and subsequent 1% pay cap that has led to a pay cut in real terms.[14]

Our previous research demonstrated an increase in poor mental health in men but not women following the Great Recession, with no clear evidence for an increase in socioeconomic inequalities.[19]

We aimed for the first time to investigate trends in both gender inequalities and socioeconomic inequalities in poor mental health in the UK following the onset of austerity, and compare these to the immediate aftermath of the 2008 recession.

## METHODS
### Dataset
Following our previous approach, we used the Health Survey for England (HSE; 1991–2014), a multistage stratified random sample designed to be nationally and regionally representative, to construct a repeat cross-sectional dataset. Details of the HSE have been published elsewhere.[20] Response levels have fallen over time but plateaued recently, remaining reasonably high at 62% in 2014 compared with 64% in 2007.[21] Weights for non-response were available from 2003. The rationale for choosing this dataset was the lengthy time period over which it has run using standardised methods, allowing consideration of very long-term trends.

### Population
The HSE general population samples were used for all analyses, restricted to those between 25 and 64 years of age to minimise misclassification of employment status among students. Those missing data on age, gender, measure of socioeconomic position (SEP), employment status or outcome were excluded. Sensitivity analysis was performed using the population aged 25–59 years

to ensure inclusion of early retirees was not impacting results.

From 1991 to 2014, there were 128 003 potential participants. A total of 7774 participants (6.1%) missing outcome data, 109 (0.1%) missing educational attainment, 2964 (2.3%) with foreign or other qualifications which could not be categorised and 37 (0.03%) missing employment status were excluded, leaving 117 119 participants (91.5%) for inclusion. For analysis using area-level deprivation from 2001 onwards where there were 73 682 potential participants, 5317 participants (7.2%) missing outcome data, 562 participants (0.7%) missing deprivation score and 25 (0.03%) missing employment status were excluded, leaving 67 778 participants for inclusion (92.0%).

### Exposure measurement and covariates
The SEP exposure measures considered were educational attainment and area-level deprivation. Highest educational attainment was available for all years except 1995 and 1996, coded into four categories: degree level or equivalent, A-level or equivalent, General Certificate of Secondary Education or equivalent and no formal qualifications. A marker of small area-level deprivation based on postal code (Index of Multiple Deprivation, IMD scored in quintiles) was available from 2001.

Covariates considered were employment status and total household income. Employment status was recorded as self-reported activity within the preceding week, coded in six categories: in employment, unemployed, retired through ill health, retired, looking after home or in education. Total household income was available from 1997, coded into quintiles.

The UK economy did not enter recession until the last quarter of 2008 (defined by two successive quarters of negative growth in GDP),[22 23] and while austerity policies were announced in mid-2010,[14] it is unlikely that health consequences would have manifested within this year. We, therefore, defined in advance all years up to and including 2008 'prerecession', the years 2009 and 2010 the 'recession period' and from 2012 onwards the 'austerity period' (outcome data were unavailable for 2011).

### Outcome measurement
Poor mental health was assessed using the General Health Questionnaire-12 (GHQ-12), a validated screening tool for common mental health problems used widely in epidemiological research, which scores self-reported symptoms of anxiety and depression.[24] The GHQ-12 formed part of the core questions in each sweep of the HSE except 1996 and 2007, though from 2010 has only been included every second year. A GHQ-12 score of 4 or greater indicates a strong likelihood of a common mental disorder,[25] and therefore defined a 'case'.

### Statistical analysis
Directly age-standardised prevalence estimates of GHQ caseness were calculated for each year, stratified by gender

and both measures of SEP. The 2013 WHO European Standard Population was used for all direct standardisations, and estimates were displayed graphically.

To quantify any potential impact of the recession and austerity on mental health by gender, multivariable logistic regression modelling was performed. First, data from each year were regressed separately to determine long-term trends in the difference between male and female caseness, adjusting for age, education and employment status. In a combined dataset of all years, models for men and women separately were then created using 2008 as the baseline/prerecession year, and adjusted for age, SEP, employment status and total household income. As the main time period of interest was following the point at which IMD was recorded routinely, we focused on this as the primary measure of SEP, given marked changes in the distribution of educational attainment over the study period. In addition to ORs, adjusted prevalence differences were derived from the logistic regression models to give a measure of change on the absolute scale.

Long-term trends in socioeconomic inequalities in mental health over time were analysed using the Relative Index of Inequality (RII), a regression-based index comparing the prevalence of the outcome between those of the theoretically lowest and highest SEP, thus giving a relative measure that could be used to draw comparisons irrespective of changes in group composition over time.[26] Analysis was performed using both SEP measures. Participants were ranked according to the chosen measure of SEP within the datasets for each individual year, with tied participants receiving the same rank; these ranks were then divided by the sample size, scaling the rank value

to between 0 and 1 with a mean of 0.5.[27] Poisson regression was used to generate prevalence risk ratios with 95% CIs,[28] comparing the most deprived with the least deprived group, which were then plotted to view trends. All models were adjusted for age and sex.

### Patient and public involvement

There was no patient or public involvement in the design of this study.

### RESULTS

Characteristics of included individuals are displayed in online supplementary appendix 1. Over the study period, there was a marked increase in women reporting degree-level education, and for both genders the number reporting no formal qualifications fell. During the main time period of interest (2005 onwards), there was little change in gender distribution.

### Mental health trends by gender

The prevalence of GHQ caseness was consistently higher in women than men over the study period (figure 1). There were three clear points of deviation from secular trends for both genders: the late 1990s, early 2000s and 2008 onwards. These deviations coincide with periods of macroeconomic disruption. During the former two time periods, the UK economy declined but avoided entering recession[22 23]; the increases in prevalence which coincide with these were patterned similarly between genders. Conversely, in 2009 following the Great Recession, there was a marked increase in age-standardised GHQ caseness

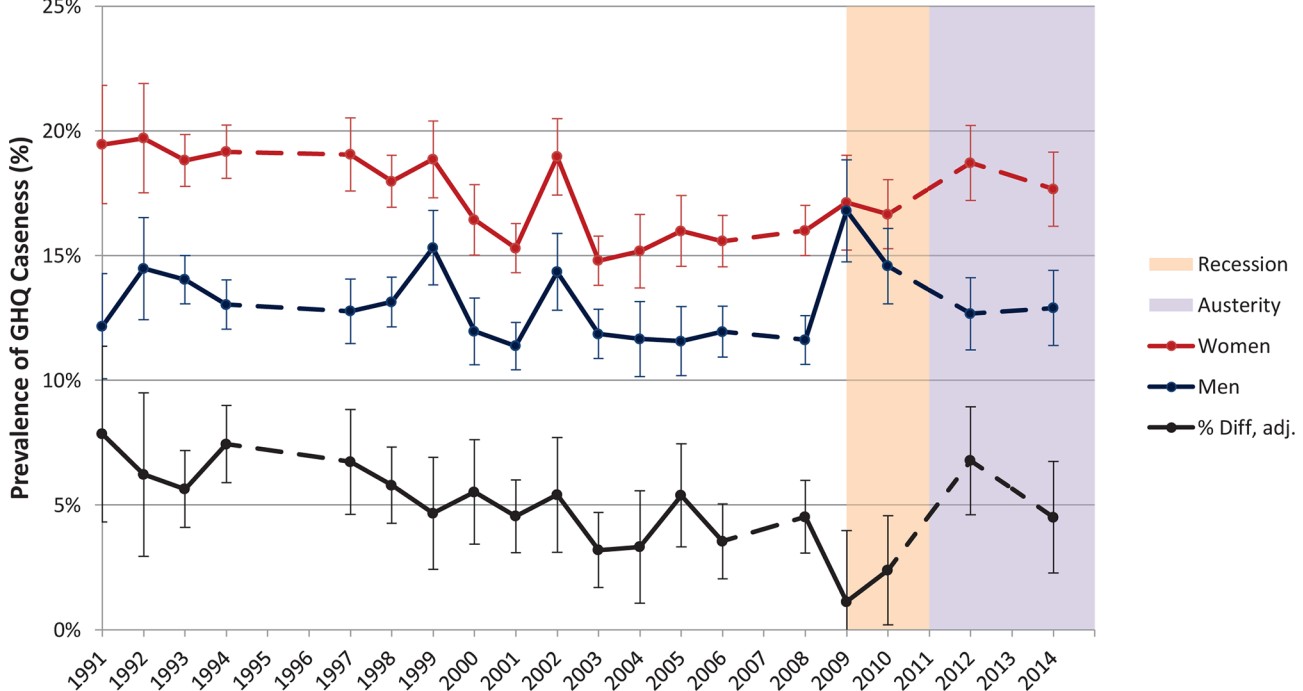

**Figure 1** Age-standardised General Health Questionnaire (GHQ) caseness in men and women aged 25–64 years, 1991–2014, with 95% CIs; and percentage point difference between male and female GHQ caseness with 95% CIs, adjusted for age, education and employment status using logistic regression. Dashed lines indicate missing years.

in men and a more modest increase in women, with only a slight improvement for men in 2010. During the austerity period this patterning altered. While in 2012 male GHQ caseness continued to decrease, female GHQ caseness increased to 18.7% (95% CI 17.2% to 20.2%), its highest observed value since 2002.

Between 1991 and 2004, the difference between male and female prevalence, adjusted for age, education and employment status, narrowed from 7.9% (95% CI 4.3% to 11.4%) to 3.3% (95% CI 1.1% to 5.6%). Despite a marked further narrowing of this gender gap in the recession period with a fall to 1.1% (95% CI −1.8% to 4.0%) in 2009 (secondary to the more marked increase in GHQ caseness for men), by 2012 it had sharply widened again to 6.8% (95% CI 4.6% to 8.9%), the largest adjusted difference between male and female prevalence since 1994. Values for all years are provided in online supplementary appendices 2 and 3.

Table 1 presents logistic regression models from the period of interest for each gender, with tables for the whole time period 2001–2014 available in online supplementary appendix 4.

GHQ caseness in men was higher in both 2009 and 2010 after adjusting for age and IMD, with prevalence predicted to have increased in the population by 5.9% (95% CI 3.2% to 8.5%, p<0.001) percentage points from 2008 to 2009. This increase remained largely unchanged (4.6%, 95% CI 2.1% to 7.1%, p<0.001) after adjustment for the potential mediating effect of employment status and household income. For men, there was no evidence of significant worsening of population mental health in either 2012 or 2014 when compared with 2008 in any model.

For women, after adjusting for age and IMD there was no evidence of an increase in GHQ caseness during the recession period. However, in 2012, the predicted increase in the population compared with 2008 was 3.0% (95% CI 1.0% to 5.1%, p=0.004), and after further adjustment for employment status and household income this remained largely unchanged at 3.1% (95% CI 1.1% to 5.1%, p=0.002). There was a smaller adjusted increase in 2014 compared with 2008 of 1.5% (95% CI −0.5% to 3.4%, p=0.142).

### Mental health trends by socioeconomic position

There was a clear socioeconomic gradient in GHQ caseness throughout the study period (figure 2). The absolute difference between the most and least deprived quintiles was among the highest recorded during the austerity period (13.5% in 2012, 11.2% in 2014) compared with smaller differences during the recession period (9.2% in 2009, 8.6% in 2010). All values are provided in online supplementary appendix 5.

Stratification by highest educational attainment produced similar trends during the recession and austerity periods (figure 3), with the exception of those with no formal qualifications. This group experienced worsening of GHQ caseness throughout the study period,

rising from 15.7% (95% CI 12.8% to 18.5%) in 1991 to 23.7% (95% CI 20.0% to 27.4%) by 2014 without seeing the recovery experienced by other groups during the austerity period. All values are provided in online supplementary appendix 6.

To explore the marked worsening for the least educated during the austerity period, further stratification by gender was performed for the period 2012–2014 : the increase in this group was predominantly among men, with age-standardised prevalence rising from 16.0% (95% CI 11.5% to 20.5%) in 2012 to 22.8% (95% CI 17.3% to 28.3%) in 2014, while for women the increase was smaller from 22.7% (95% CI 17.9% to 27.5%) in 2012 to 24.7% (95% CI 19.8% to 29.5%) in 2014.

Relative socioeconomic inequalities in GHQ caseness have been consistently observed since 1999 (figure 4). Inequalities in GHQ caseness have increased from the late 1990s to the immediate prerecession period, with inequalities generally larger by area-level deprivation. During the recession period, there was a slight reduction in socioeconomic inequalities, with RII by education falling from 1.8 (95% CI 1.5 to 2.2, p<0.001) in 2008 to 1.6 (95% CI 1.2 to 2.1, p=0.001) in 2010 and by IMD quintile from 2.3 (95% CI 1.9 to 2.8, p<0.001) in 2008 to 1.9 (95% CI 1.4 to 2.4, p<0.001) in 2010. However, these trends reversed during the austerity period, and by 2014 both RIIs had returned to prerecession levels. All values are provided in online supplementary appendix 7.

For all analyses, sensitivity analysis excluding those aged 60–64 years did not affect trends.

### DISCUSSION

In this large repeat cross-sectional study of a representative sample of the English population, we found mental health worsened for women following the onset of austerity policies, while men saw a recovery to prerecession levels. As a result of the changes, gender inequalities in poor mental health widened during the austerity period, reversing the trend from 1991 to 2004 of gradual improvement. We also found that socioeconomic inequalities in poor mental health narrowed in the immediate years following the 2008 recession but widened during the austerity period. While it is not possible to draw definitive causal conclusions from this study, our findings are useful in examining changes in secular trends and their chronological association with macroeconomic events and policies.

There is conflicting evidence in existing literature around whether mental health inequalities by gender or socioeconomic position have widened in the UK since the recession. Our previous work suggested males saw the sharpest worsening of mental health, and found no evidence of widening socioeconomic inequalities when existing trends were taken into account.[19] However, this was prior to the onset of austerity. More recent evidence showed a more marked worsening of mental health for women in 2014 compared with 2007, but did not take into

**Table 1** Multiple logistic regression models (with ORs and % point difference) for participants of each gender, 2005–2014 (2008 as prerecession reference year)

### Regression models for men (n=24 930)

| Year | Model 1: adjusted for age, IMD | | | | Model 2: adjusted for age, IMD, employment | | | | Model 3: adjusted for age, IMD, employment, income | | | |
|---|---|---|---|---|---|---|---|---|---|---|---|---|
| | OR | P values | Lower 95% CI | Upper 95% CI | OR | P values | Lower 95% CI | Upper 95% CI | OR | P values | Lower 95% CI | Upper 95% CI |
| 2005 | 1.03 | 0.722 | 0.86 | 1.24 | 0.92 | 0.400 | 0.76 | 1.12 | 0.90 | 0.303 | 0.74 | 1.10 |
| 2006 | 1.06 | 0.476 | 0.91 | 1.24 | 1.04 | 0.676 | 0.88 | 1.22 | 1.03 | 0.765 | 0.87 | 1.21 |
| 2008 | 1.00 | – | – | – | 1.00 | – | – | – | 1.00 | – | – | – |
| 2009 | 1.64 | <0.001 | 1.34 | 2.00 | 1.55 | <0.001 | 1.25 | 1.93 | 1.53 | <0.001 | 1.24 | 1.91 |
| 2010 | 1.28 | 0.009 | 1.06 | 1.53 | 1.26 | 0.021 | 1.04 | 1.52 | 1.26 | 0.018 | 1.04 | 1.53 |
| 2012 | 1.15 | 0.147 | 0.95 | 1.38 | 1.10 | 0.340 | 0.91 | 1.33 | 1.10 | 0.342 | 0.91 | 1.33 |
| 2014 | 1.13 | 0.215 | 0.93 | 1.37 | 1.17 | 0.126 | 0.96 | 1.43 | 1.18 | 0.108 | 0.96 | 1.44 |
| Year | % Diff. | P values | Lower 95% CI | Upper 95% CI | % Diff. | P values | Lower 95% CI | Upper 95% CI | % Diff. | P values | Lower 95% CI | Upper 95% CI |
| 2005 | 0.34 | 0.723 | −1.53 | 2.20 | −0.75 | 0.396 | −2.50 | 0.99 | −0.93 | 0.298 | −2.67 | 0.82 |
| 2006 | 0.58 | 0.477 | −1.01 | 2.17 | 0.33 | 0.676 | −1.21 | 1.86 | 0.23 | 0.765 | −1.30 | 1.77 |
| 2008 | 0.00 | – | – | – | 0.00 | – | – | – | 0.00 | – | – | – |
| 2009 | 5.88 | <0.001 | 3.24 | 8.52 | 4.72 | <0.001 | 2.20 | 7.23 | 4.62 | <0.001 | 2.11 | 7.12 |
| 2010 | 2.67 | 0.012 | 0.60 | 4.73 | 2.29 | 0.024 | 0.30 | 4.28 | 2.37 | 0.021 | 0.36 | 4.38 |
| 2012 | 1.44 | 0.154 | −0.54 | 3.42 | 0.89 | 0.344 | −0.95 | 2.73 | 0.89 | 0.346 | −0.96 | 2.74 |
| 2014 | 1.26 | 0.223 | −0.77 | 3.30 | 1.53 | 0.134 | −0.47 | 3.52 | 1.63 | 0.115 | −0.40 | 3.65 |

### Regression models for women (n=31 413)

| Year | Model 1: adjusted for age, IMD | | | | Model 2: adjusted for age, IMD, employment | | | | Model 3: adjusted for age, IMD, employment, income | | | |
|---|---|---|---|---|---|---|---|---|---|---|---|---|
| | OR | P values | Lower 95% CI | Upper 95% CI | OR | P values | Lower 95% CI | Upper 95% CI | OR | P values | Lower 95% CI | Upper 95% CI |
| 2005 | 1.06 | 0.426 | 0.92 | 1.22 | 1.05 | 0.531 | 0.91 | 1.21 | 1.02 | 0.762 | 0.88 | 1.18 |
| 2006 | 0.91 | 0.153 | 0.81 | 1.03 | 0.90 | 0.102 | 0.80 | 1.02 | 0.89 | 0.077 | 0.79 | 1.01 |
| 2008 | 1.00 | – | – | – | 1.00 | – | – | – | 1.00 | – | – | – |
| 2009 | 1.06 | 0.537 | 0.89 | 1.25 | 1.07 | 0.429 | 0.90 | 1.28 | 1.07 | 0.431 | 0.90 | 1.28 |
| 2010 | 1.00 | 0.980 | 0.86 | 1.15 | 0.99 | 0.849 | 0.85 | 1.14 | 0.99 | 0.922 | 0.86 | 1.15 |
| 2012 | 1.24 | 0.003 | 1.08 | 1.42 | 1.24 | 0.003 | 1.07 | 1.43 | 1.25 | 0.002 | 1.09 | 1.45 |
| 2014 | 1.10 | 0.208 | 0.95 | 1.27 | 1.11 | 0.171 | 0.96 | 1.28 | 1.12 | 0.138 | 0.97 | 1.29 |
| Year | % Diff. | P values | Lower 95% CI | Upper 95% CI | % Diff. | P values | Lower 95% CI | Upper 95% CI | % Diff. | P values | Lower 95% CI | Upper 95% CI |
| 2005 | 0.79 | 0.429 | −1.16 | 2.74 | 0.60 | 0.533 | −1.29 | 2.50 | 0.29 | 0.762 | −1.60 | 2.18 |
| 2006 | −1.16 | 0.153 | −2.74 | 0.43 | −1.28 | 0.102 | −2.82 | 0.25 | −1.39 | 0.077 | −2.94 | 0.15 |
| 2008 | 0.00 | – | – | – | 0.00 | – | – | – | 0.00 | – | – | – |
| 2009 | 0.73 | 0.541 | −1.61 | 3.07 | 0.93 | 0.435 | −1.40 | 3.25 | 0.93 | 0.436 | −1.41 | 3.27 |
| 2010 | −0.02 | 0.98 | −1.96 | 1.91 | −0.18 | 0.848 | −2.07 | 1.70 | −0.10 | 0.922 | −2.00 | 1.81 |
| 2012 | 3.04 | 0.004 | 0.99 | 5.08 | 2.90 | 0.004 | 0.93 | 4.88 | 3.11 | 0.002 | 1.11 | 5.11 |
| 2014 | 1.27 | 0.213 | −0.73 | 3.28 | 1.34 | 0.176 | −0.60 | 3.29 | 1.47 | 0.142 | −0.49 | 3.44 |

IMD, Index of Multiple Deprivation.

account intervening years.[29] A large study of pan-European data including the UK found no systematic influence of the recession on socioeconomic inequalities in depression up to 2014,[30] but did not differentiate between the immediate recessionary period and the period following any economic policy response. Work by

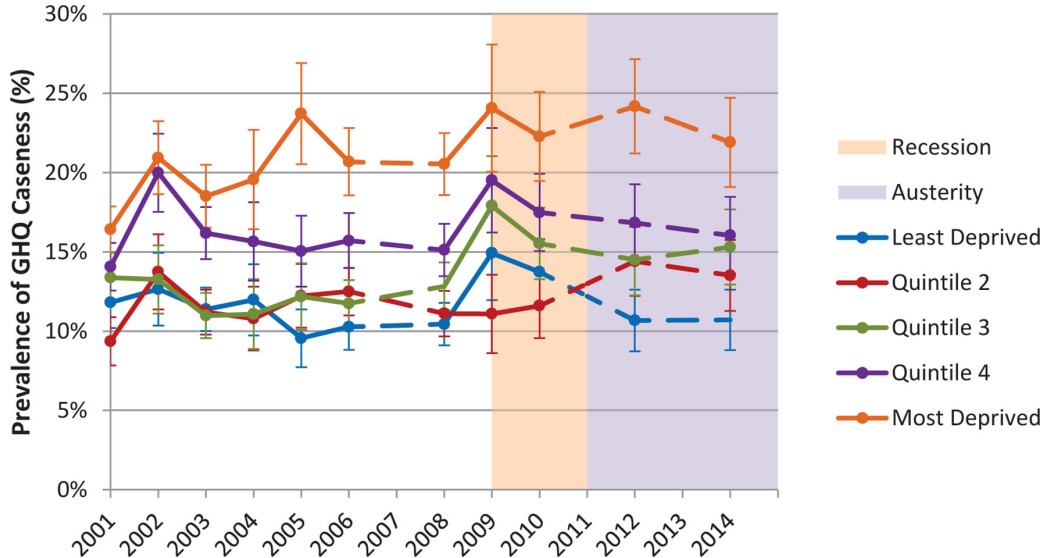

**Figure 2** Age-sex standardised General Health Questionnaire (GHQ) caseness by Index of Multiple Deprivation quintile in 25–64 year olds, 2001–2014, with 95% CIs. Dashed lines indicate missing years.

Barr *et al* suggested that from 2009 to 2013 there may have been a widening of socioeconomic inequalities in mental health in the UK.[31] However, this used self-reported diagnoses and only two broad categories of socioeconomic group. Our study adds clarity to both areas.

There is no consensus around what factors are responsible for the gender gap in poor mental health. There is little evidence it results from purely genetic or biological differences, with sociocultural roles, adverse life events and learnt psychological attributes thought more likely contributing factors.[32] Our findings of a reversal in trend direction echo those of others who have begun to raise concerns about the mental health of UK women in recent

years, particularly young women.[29 33] The timing of this reversal in relation to austerity reforms and the differential gender patterning of austerity[17] could indicate that the change for women may be secondary to the policy response rather than the economic crisis itself—particularly, as evidence emerges of likely adverse impacts of specific policy reforms affecting women, such as restrictions to income support being linked to deteriorations in mental health among lone parents.[34]

The finding of a reversal in trend towards widening socioeconomic inequalities following the onset of austerity adds to the evidence base arguing such measures may mediate the link between macroeconomic change

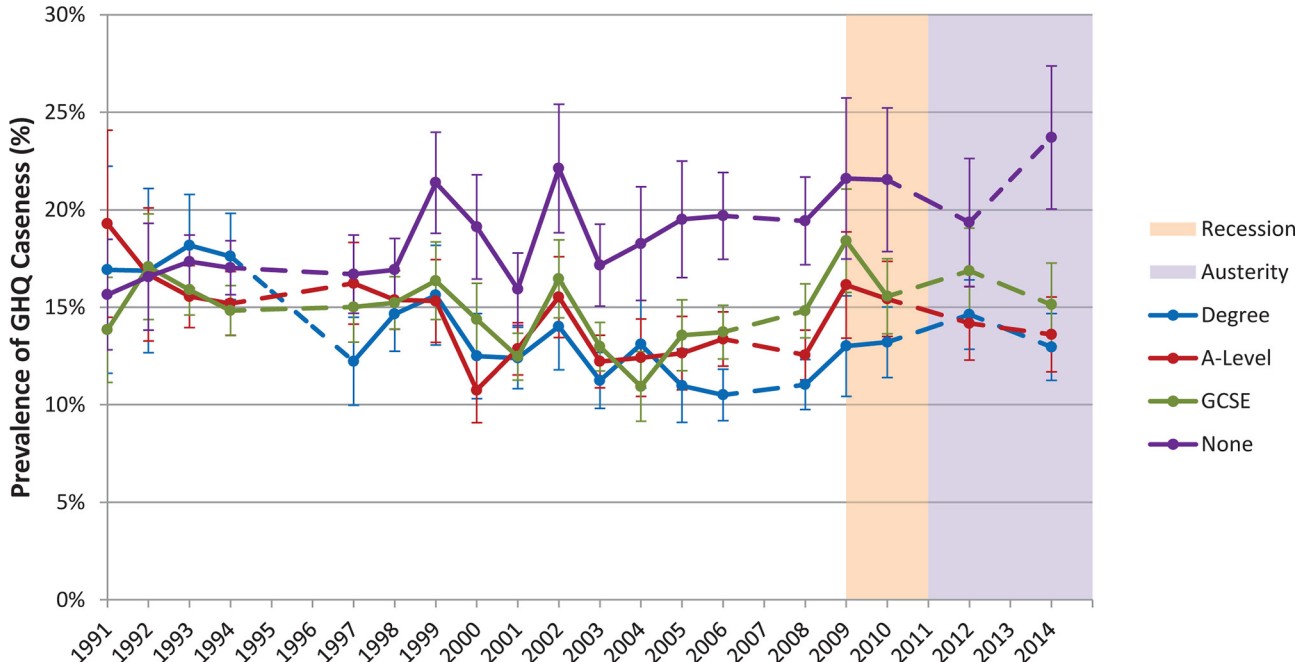

**Figure 3** Age-sex standardised General Health Questionnaire (GHQ) caseness by education level in 25–64 year olds, 1991–2014, with 95% CIs. Dashed lines indicate missing years. GCSE, General Certificate of Secondary Education.

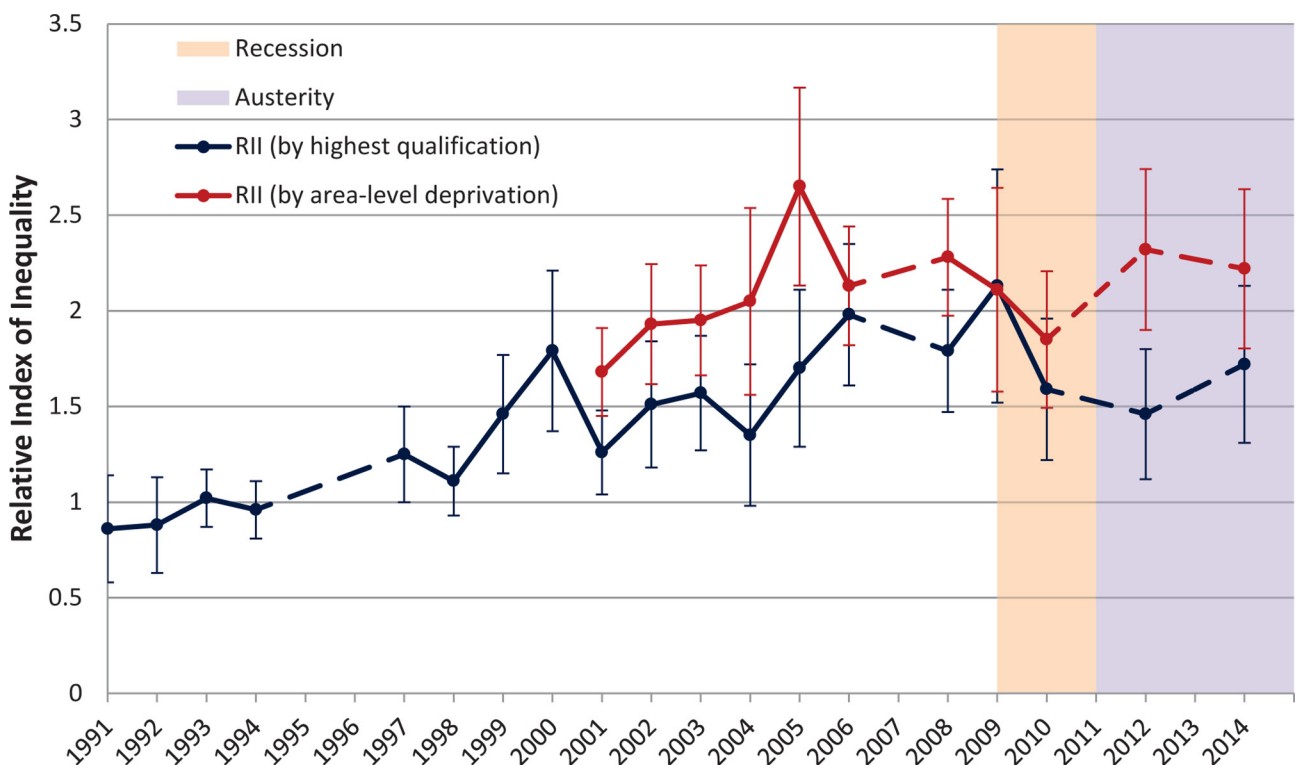

**Figure 4** Relative Index of Inequality (RII) in General Health Questionnaire caseness in 25–64 year olds by education level and Index of Multiple Deprivation quintile, 1991–2014, with 95% CIs. Dashed lines indicate missing years.

and mental health.[2 10] Ecological studies using pan-European data suggest that the direct effect of unemployment on suicide rates was greater in countries with lower social spending,[35] and conversely, higher government spending on unemployment support may mitigate adverse impacts on self-rated health.[36] On a relative scale the widening of socioeconomic inequalities postausterity is small in the context of long-term trends, particularly by highest qualification (figure 4), and the degree and timing of short-term trends around the recession and austerity period also differ between measures of SEP, possibly explaining the current lack of consensus in the literature.[30 31]

The marked divergence for those with no formal qualifications by 2014 may support the hypothesis that those in low-skilled jobs (who are known to experience poorer health outcomes[37]) may be worst affected by reduced in-work financial support or worsening job conditions such as increased insecure work.[38] Their divergence may also be partly attributable to changes in demographics over the study period, with the group achieving no qualifications becoming smaller and more homogeneous over time. Regardless, they are notable outliers in 2014, identifying this group as particularly high risk for poor mental health.

### Strengths and limitations

Our study has a number of important strengths. The HSE is a large, nationally and regionally representative survey which has used standard protocols over a long period of time. The GHQ-12 is a validated and commonly used measure, and outcome data were available for most years

allowing detailed consideration of trends. While there is some debate about the most appropriate threshold to use to determine caseness in different populations,[39] we chose a cut-off value that has been used previously with this population[19 40] and which indicates a strong likelihood of common mental disorder,[25] increasing specificity and reducing the likelihood of false positive cases. The use and comparison of two measures of SEP is useful in demonstrating consistency of trends between SEP and poor mental health.

Our study also has some limitations which must be considered. The use of cross-sectional rather than individual longitudinal data mean the ability to derive causal inferences is limited; however, it does overcome attrition bias in cohort studies which can commonly lead to an underestimation of inequalities.[41] As data were not collected on whether individuals were subject to specific austerity measures, this could not be included as an explanatory variable. Household income was felt to be a reasonable proxy given that most reforms were associated with financial loss.[16] It is acknowledged that the impact of an economic crisis or subsequent policies is not necessarily immediate and is likely to be mediated by related factors such as long-term unemployment. It is therefore possible that trends in GHQ caseness may have been influenced by other factors apart from austerity, including observed trends reflecting the longer term impacts of earlier macroeconomic exposures. Finally, it is unfortunate that outcome data were not available from 2007, 2011, 2013 or 2015, as this would have strengthened the evidence for the assessment of trends.

Further research using longitudinal data would add strength to any argument for causality, as would replication using alternative outcome measures, such as antidepressant prescriptions. Distinguishing between the impact of different components of austerity measures, for example, public sector employment terms, welfare reforms or access to community services could add further nuance to our reporting of their potential combined impact and overcome this identified limitation. Furthermore, increasing devolution provides the opportunity to study differences in policy approaches within the UK.[42] Cross-national comparisons would also be useful in determining whether observed trends are replicated elsewhere, and whether impacts are dependent on levels of austerity, and natural experiment approaches could strengthen causal inference.[43] Finally, it is clearly important to see whether the observed trajectories in mental health inequalities have continued following 2014, particularly given that more severe welfare reforms were initiated in 2015.[16]

## CONCLUSIONS

This study adds to what the European Psychiatric Association in 2016 described as an emerging 'broad consensus about the deleterious consequences of economic crises on mental health'.[44] The gender gap in mental health, which had been improving prior to the recession, appears to be sharply widening again following the onset of austerity policies which have largely focused on women. Those in the most deprived groups have been shown to be at potentially heightened risk of poor mental health following the onset of austerity, with the least educated at highest risk.

These findings are alarming, particularly given that since the time period studied there have been further cuts to mental health provision which mean the issue may now be worse.[45] Labonté and Stuckler argue in strong terms that, based on current evidence of economic, health and social harms, austerity policies threaten to 'imperil the world's population' without radical reform.[2] Policy-makers in the UK and those considering embarking on or continuing austerity measures elsewhere in the world should be aware that these may have adverse health impacts for their populations.

**Acknowledgements** We would like to thank Frank Popham for providing feedback on drafts of the manuscript. We are extremely grateful to the individuals who took part in all cycles of the Health Survey for England, and all those involved in its administration over the years. We would like to acknowledge the assistance of the UK Data Archive for providing access to the data; the information centre for health and social care and department of health for sponsoring the Health Survey for England and the principal investigators of the Health Survey for England, Natcen Social Research and the Department of Epidemiology and Public Health at the Royal Free and University College Medical School. The responsibility for the analysis presented here lies solely with the authors.

**Contributors** RMT serves as guarantor for this article. RMT and SVK conceived the idea for the study. RMT performed the statistical analysis and wrote the first draft of the article. CLN and SVK assisted in research design, interpretation of findings and critical revision of the manuscript.

**Funding** SVK is funded by an NRS Senior Clinical Fellowship (SCAF/15/02), the Medical Research Council (MC_UU_12017/13 and MC_UU_12017/15) and Scottish Government Chief Scientist Office (SPHSU13 and SPHSU15). CLN is funded by the Medical Research Council (MR/R024774/1).

**Competing interests** None declared.

**Ethics approval** Ethical approval was not required for this study as it used previously collected data. Ethical approval for each year of the survey was obtained by the Health Survey for England team.

**Provenance and peer review** Not commissioned; externally peer reviewed.

**Data sharing statement** No additional data are available.

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
