## [Reviewer comments · BMJ Open]

ARTICLE DETAILS

TITLE (PROVISIONAL)	Trends in gender and socioeconomic inequalities in mental health following the Great Recession and subsequent austerity policies: a repeat cross-sectional analysis of the Health Surveys for England
AUTHORS	Thomson, Rachel; Niedzwiedz, Claire; Katikireddi, Srinivasa

VERSION 1 – REVIEW

REVIEWER	Ian Colman University of Ottawa, Canada
REVIEW RETURNED	12-Apr-2018

GENERAL COMMENTS	This paper sought to investigate trends in gender and socioeconomic inequalities in poor mental health from 1991 to 2014. The authors used data from a repeated cross-sectional survey, and were particularly interested in the post-recession and introduction to austerity policies periods. The authors concluded that inequalities in poor mental health narrowed post-recession, but increased following austerity measures. While the results are very interesting and potentially policy relevant, the authors may be over-interpreting the findings. Specific comments: 1. The language used throughout indicates that the authors strongly believe that the recession and subsequent austerity measures caused the observed changes in inequalities with regards to mental health. However, while the results are compelling, the trends observed could be attributable to many other secular changes over time. I recommend the authors soften the causal language throughout the manuscript, and consider other possible changes among variables that were not assessed in the surveys.2. With regards to the above, Figure 4 suggests that the results are not as consistent with the proposed hypothesis as is suggested. While the long-term trends in inequalities are similar according to both education and area-level deprivation, the short-term trends around the times of the recession and austerity measures appear to differ. Notably, there is no apparent increase in inequality by area level deprivation from 2010 to 2014, which is not consistent with the authors' conclusion. This bears a comment in the Discussion, at the very least.3. It's very difficult to interpret the key findings with respect to inequality, given that the measure of inequality is described in very brief terms in the Methods section. Given how fundamental this measure is to the interpretation of the findings, much more detail is needed. Currently, there is only one sentence and it provides very little meaningful information.
--

REVIEWER	Luis Rajmil Barcelona Spain
REVIEW RETURNED	24-Apr-2018

GENERAL COMMENTS	The present study includes data from the Health Survey for England 1991-2014, and shows inequalities in mental health according to gender and socioeconomic status. Authors attempt to differentiate the impact of the economic crisis and the government's austerity responses. The impact of the crisis showed a significant increase in GHQ-12 scores between 2008 and 2009 in men and an increase was observed in women in 2012. Those of lower socioeconomic level had the most important impact during the entire study period. The following aspects could be taken into account to try to improve the presentation of the study: 1) The GHQ is a universally extended measure, although not fully consensual in terms of what the measure attempts to collect. In fact, just as the authors comment, it only collects symptoms of anxiety and depression. In fact, Goldberg himself had already published some limitations of the instrument (i.e. the performance of the cut-off point depends on the population to which it is addressed: Goldberg DP, Oldehinkel T, Ormel J. Why GHQ threshold varies from one place to another. Psychol Med. 1998; 28: 915-21). Perhaps this aspect could be added and authors could justify why they used 3/4 cut-off point. 2) It should be clarified if the 95%CI are represented in the figures. The results are presented as annual or time period trends but to know if changes are significant, it should be analyzed if changes in time trends are significant, or alternatively 95% CI not overlap. 3) The interpretation of changes between 2008 and 2009 might be better explained if it is made clear that the GHQ is much more worsening in men than in women. It could be interpreted as if the decrease in differences represents an improvement for women. 4) Another limitation that could be deepened is that the impact of the crisis and / or austerity on mental health is not necessarily immediate. The authors mention this fact but perhaps it could be more explicit that a period of time can pass between exposure to crisis, austerity measures, and worsening mental health. For example, long-term unemployment surely has an impact on mental health and can generate more impact when social benefits are cut.
--

VERSION 1 – AUTHOR RESPONSE

Reviewer(s)' Comments to Author:

Reviewer #1

This paper sought to investigate trends in gender and socioeconomic inequalities in poor mental health from 1991 to 2014. The authors used data from a repeated cross-sectional survey, and were particularly interested in the post-recession and introduction to austerity policies periods. The authors concluded that inequalities in poor mental health narrowed post-recession, but increased following austerity measures. While the results are very interesting and potentially policy relevant, the authors may be over-interpreting the findings.

Specific comments:

1. The language used throughout indicates that the authors strongly believe that the recession and subsequent austerity measures caused the observed changes in inequalities with regards to mental health. However, while the results are compelling, the trends observed could be attributable to many other secular changes over time. I recommend the authors soften the causal language throughout the manuscript, and consider other possible changes among variables that were not assessed in the surveys.

Language has been edited throughout to reduce any direct reference to 'impact' of recession/austerity – instead trends are presented as occurring during pre-recession period, recession period or austerity period without commenting on causality. We have made additional reference in the Discussion to the inability to determine causality due to the cross-sectional study design, and the possibility of confounding due to other secular changes has been included as a Limitation. For example:

“While it is not possible to draw definitive causal conclusions from this study, our findings are useful in examining changes in secular trends and their chronological association with macroeconomic events and policies.” (page 12 line 7-12)

2. With regards to the above, Figure 4 suggests that the results are not as consistent with the proposed hypothesis as is suggested. While the long-term trends in inequalities are similar according to both education and area-level deprivation, the short-term trends around the times of the recession and austerity measures appear to differ. Notably, there is no apparent increase in inequality by area level deprivation from 2010 to 2014, which is not consistent with the authors' conclusion. This bears a comment in the Discussion, at the very least.

We believe the reviewer referred to a perceived lack of change in the Relative Index of Inequality (RII) by highest qualification between 2010 and 2014, rather than area-level deprivation. We have added a comment on this in the Discussion, in addition to a comment on the differences between the results by both measures:

“On a relative scale the widening of socioeconomic inequalities post-austerity is small in the context of long-term trends, particularly by highest qualification (Figure 4), and the degree and timing of short-term trends around the recession and austerity period also differ between measures of SEP, possibly explaining the current lack of consensus in the literature.” (page 13 line 7-10)

3. It's very difficult to interpret the key findings with respect to inequality, given that the measure of inequality is described in very brief terms in the Methods section. Given how fundamental this measure is to the interpretation of the findings, much more detail is needed. Currently, there is only one sentence and it provides very little meaningful information.

We have now added more detailed information about the RII and its calculation to the Methods section:

“Long-term trends in socioeconomic inequalities in mental health over time were analysed using the relative index of inequality (RII), a regression-based index comparing the prevalence of the outcome between those of the theoretically lowest and highest SEP, thus giving a relative measure that could be used to draw comparisons irrespective of changes in group composition over time. Analysis was performed using both SEP measures. Participants were ranked according to the chosen measure of SEP within the datasets for each individual year, with tied participants receiving the same rank; these ranks were then divided by the sample size, scaling the rank value to between 0 and 1 with a mean of 0.5. Poisson regression was used to generate prevalence risk ratios with 95% confidence intervals, comparing the most deprived with the least deprived group, which were then plotted to view trends.” (page 6 line 32-page 7 line 7)

Reviewer #2

The present study includes data from the Health Survey for England 1991-2014, and shows inequalities in mental health according to gender and socioeconomic status. Authors attempt to differentiate the impact of the economic crisis and the government's austerity responses. The impact of the crisis showed a significant increase in GHQ-12 scores between 2008 and 2009 in men and an increase was observed in women in 2012. Those of lower socioeconomic level had the most important impact during the entire study period.

The following aspects could be taken into account to try to improve the presentation of the study:

1) The GHQ is a universally extended measure, although not fully consensual in terms of what the measure attempts to collect. In fact, just as the authors comment, it only collects symptoms of anxiety and depression. In fact, Goldberg himself had already published some limitations of the instrument (i.e. the performance of the cut-off point depends on the population to which it is addressed: Goldberg DP, Oldehinkel T, Ormel J. Why GHQ threshold varies from one place to another. *Psychol Med.* 1998; 28: 915-21). Perhaps this aspect could be added and authors could justify why they used 3/4 cut-off point.

We have now added reference to this paper and included justification of the choice of cut-off in the Discussion when describing the strengths and limitations of the study:

“While there is some debate about the most appropriate threshold to use to determine caseness in different populations, we chose a cut-off value that has been used previously with this population and which indicates a strong likelihood of common mental disorder, increasing specificity and reducing the likelihood of false positive cases.” (page 13 line 22-25)

2) It should be clarified if the 95%CI are represented in the figures. The results are presented as annual or time period trends but to know if changes are significant, it should be analyzed if changes in time trends are significant, or alternatively 95% CI not overlap.

Figure legends have been added to the end of the manuscript clarifying that error bars represent 95% confidence intervals. Changes in time trends are tested statistically using logistic regression, with all values displayed in Table 1, for each gender. While between-year changes are not tested statistically for analysis by socioeconomic position, all point estimates for the RII including 95% confidence intervals are presented as tables in the Supplementary Materials, with 95% confidence intervals also reported for relevant figures in the main text of the Results section.

3) The interpretation of changes between 2008 and 2009 might be better explained if it is made clear that the GHQ is much more worsening in men than in women. It could be interpreted as if the decrease in differences represents an improvement for women.

A comment clarifying this has been added in the Results section when discussing the change in % difference by gender between 2008 and 2009. As we have commented on the marked worsening in GHQ caseness for men in 2009 and possible explanations for this in our previous work^[1] and the main focus of this paper was how trends have changed since the onset of austerity measures in 2010, we have chosen not to further comment on this in our Discussion.

4) Another limitation that could be deepened is that the impact of the crisis and / or austerity on mental health is not necessarily immediate. The authors mention this fact but perhaps it could be more explicit that a period of time can pass between exposure to crisis, austerity measures, and worsening mental health. For example, long-term unemployment surely has an impact on mental health and can generate more impact when social benefits are cut.

This limitation is gratefully acknowledged, and has been added to the Discussion.

Editor Comments to Author:

- Please ensure that your manuscript is formatted according to our Instructions for Authors (<http://bmjopen.bmj.com/pages/authors/>). For example, do not include the sections "What is already known on this subject" and "What this study adds".

These sections have been removed, and the rest of the manuscript checked against the suggested formatting in the Instructions for Authors.

- Please revise the title to state the research question, study design, and settings. This is the preferred format for the journal. See published articles for examples.

The title has been revised from 'Trends in gender and socioeconomic inequalities in mental health in England following the Great Recession and subsequent austerity policies' to 'Trends in gender and socioeconomic inequalities in mental health following the Great Recession and subsequent austerity policies: a repeat cross-sectional analysis of the Health Surveys for England'

- Along with your revised manuscript, please include a completed copy of the STROBE checklist indicating the line/page numbers where the relevant information can be found (www.strobe-statement.org/index.php?id=strobe-home)

A STROBE checklist has been completed and uploaded with the resubmission.

- Please include figure legends at the end of your main manuscript.

Figure legends have been added to the end of the manuscript.

Kind regards,

Dr Rachel Thomson [on behalf of all authors]

[1] Katikireddi SV, Niedzwiedz CL, Popham F. Trends in population mental health before and after the 2008 recession: a repeat cross-sectional analysis of the 1991-2010 Health Surveys of England. *BMJ Open* 2012;2(5) doi: 10.1136/bmjopen-2012-001790

VERSION 2 – REVIEW

REVIEWER	Ian Colman University of Ottawa, Canada
REVIEW RETURNED	20-Jun-2018

GENERAL COMMENTS	The authors have comprehensively responded to the reviewer concerns. The resulting paper is very interesting and should make a fine contribution to the literature.
---

REVIEWER	Luis Rajmil Retired
REVIEW RETURNED	10-Jun-2018

GENERAL COMMENTS	The revised version of the manuscript have improved and authors have answered satisfactorily to my questions and comments
---